# Small Talk, Big Impact: The Energy Cost of Thanking AI

## Abstract

Being nice doesn't cost you anything - or does it? In this paper, we quantify the energy cost of seemingly innocuous messages such as "thank you" when interacting with large language models to convey politeness. Using real-world conversation traces and fine-grained energy measurements, we quantify how input length, output length and model size affect energy use. While politeness is our motivating example, it also serves as a controlled and reproducible proxy for measuring the energy footprint of a typical LLM interaction. Our findings provide actionable insights for building more sustainable and efficient LLM applications, especially in increasingly widespread real-world contexts like chat. As user adoption grows and billions of prompts are processed daily, understanding and mitigating this cost becomes crucial - not just for efficiency, but for sustainable AI deployment.

## 1 Introduction

Politeness is an instinctive part of human conversation. From daily greetings to simple "thank you" messages, these small acts of courtesy shape how we communicate - even with machines. As large language models (LLMs) become increasingly integrated into our workflows and devices, their interactions begin to mirror human expectations, including the social norm of politeness. This has been referred to as the "ELIZA effect", named after the AI chatbot developed in the 1960s that was meant to imitate a psychotherapist based on pattern matching and rule-based programming. Despite the basic text processing and limited number of responses available to ELIZA, many of its early users became convinced of its intelligence and ability to understand their emotions, leading them to treat it with a degree of politeness and deference that were usually reserved for human interlocutors (Hofstadter, 1995). This phenomenon continues to persist to this day, with users of LLM-based chatbots and systems anthropomorphizing them and using polite language in their interactions (Chow, 2023).

LLMs process every message as a prompt in itself – a polite "thank you!" at the end of a chat is not just a gesture, but triggers a full inference pass through the billions of parameters that constitute the model. And as each one of these computations consumes non-negligible amounts of energy, this raises an important question: how can we measure the energy cost of being polite in our interactions with LLMs? We answer this question via a series of empirical experiments, prompting open-source LLMs with different approaches and measuring the amount of energy used for each response to a prompt conveying politeness. Our results emphasize that polite interactions, while seemingly innocuous, can accumulate into substantial compute overhead at scale - and provide actionable strategies to reduce their impact. While polite prompts incur measurable energy costs, they may also serve important social and functional roles - improving alignment, mitigating harmful responses, and enhancing user trust. This raises a broader trade-off between efficiency and safety in human–LLM interaction, which we aim to quantify and discuss.

Our key contributions are:

- We measure the GPU, CPU, and RAM energy consumption of generating polite replies with open-source instruction-tuned LLMs across thousands of realistic chat completions.
- We decompose energy usage into prefill and decode phases, and relate these to prompt and output length through a closed-form latency model - showing a clear alignment between empirical trends and theoretical complexity.

- We analyze how model size affects both verbosity and energy use, and show that larger models not only consume more per token, but also tend to generate longer replies.
- We release all measurement code, processed datasets, and latency models to support reproducibility. datasets and results

## 2 METHODOLOGY

### 2.1 DATASET AND EXPERIMENTAL SETUP

To evaluate the energy cost of polite interactions, we constructed a dataset of 10,000 chat-based conversations ending with a "thank you" message from the user. These were derived from the `ultrachat_200k` dataset (Ding et al., 2023) and reformatted to match the instruction-following prompt template expected by `Instruct` models, simulating real-world assistant usage scenarios. Each prompt submitted to the model included the entire conversation history up to that point, ensuring a realistic multi-turn context.

For every unique prompt, we performed 5 warmup runs to stabilize performance and cache behavior, followed by 10 measurement runs for each generation phase. Specifically, we conducted 10 *prefill-only* generations (constrained to a single output token) and 10 full generations (up to 256 output tokens). This allowed us to separately estimate the energy consumption of the *prefill* and *decode* phases, by subtracting the average prefill energy from the full generation energy. We also logged input/output lengths, latency, and energy consumption by hardware component for each run.

### 2.2 HARDWARE AND SOFTWARE ENVIRONMENT

All experiments were run on a dedicated inference server equipped with an NVIDIA H100 SXM GPU (80GB) and 8 AMD EPYC 7R13 CPU cores, with no co-scheduled jobs.

Energy was measured using:

- **GPU:** NVIDIA Management Library (NVML),
- **CPU:** `pyRAPL` (Intel RAPL counters),
- **RAM:** CodeCarbon's model-based estimation[1].

### 2.3 MODELS AND PRECISION

Our core analysis focused on the **LLaMA 3.1–8B-Instruct** model in `float32`, run via the standard Transformers library (Wolf et al., 2020). In this setting, generation used `batch size 1`. We chose **LLaMA 3.1–8B-Instruct** for its strong relevance: as of July 2025 it is the **2nd most liked** and **3rd most downloaded** model on the Hugging Face Hub, making it a representative and impactful open-source option.

To study how *model size* influences energy usage, we extended our tests beyond a single model. In particular, the **Qwen 2.5** family was selected because all models share the same architecture and tokenizer, which enables a controlled comparison focused solely on scaling effects. We complemented this with an additional widely used open-source baseline.

We therefore evaluated:

- **Qwen 2.5 family:** 0.5B, 1.5B, 3B, 7B, and 14B,
- **Mistral-7B-Instruct-v0.3**.

## 3 THE ENERGY COST OF A SIMPLE "THANK YOU"

We measured the energy required to generate a reply to a polite "thank you" message using the **LLaMA 3.1–8B-Instruct FP32** model, deployed on a single NVIDIA H100 GPU. Across 10,000 such interactions, we observed a mean total energy consumption of:

---

[1] https://mlco2.github.io/codecarbon/methodology.html#ram

- **0.202 ± 0.096 Wh** on the GPU,

- **0.024 ± 0.014 Wh** on the CPU,

- **0.019 ± 0.010 Wh** from RAM.

The total energy per polite interaction thus averages **0.245 Wh**, which is equivalent to powering a 5W LED bulb for nearly 3 minutes [2].

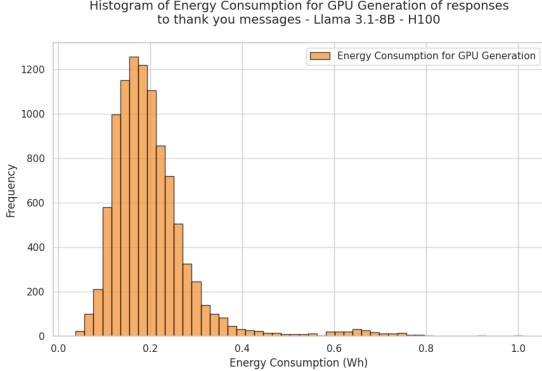

Figure 1: Distribution of GPU energy consumption across "thank you" generations. The long tail indicates variability due to prompt and output length.

**GPU usage dominates the total energy profile**, with a contribution nearly an order of magnitude larger than the CPU or RAM. The GPU also exhibits higher variance, reflecting its sensitivity to sequence length and runtime context. Figure 1 shows the distribution of GPU energy per generation, which is **right-skewed with a long tail** - indicating that some completions are disproportionately costly, especially when the model produces longer or more verbose replies. This variability is primarily driven by differences in **prompt and output length**. While "thank you" is always the last user message, the surrounding conversation history and the model's verbosity can significantly influence token count - and thus, energy consumption.

## 4 COMPONENT-WISE ENERGY CONSUMPTION

To better understand where energy is spent during LLM inference, we break down the generation process into two main phases: *prefill*, which encodes the entire prompt and generates the first token, and *decode*, which generates the remaining tokens one by one using the cached context.

As shown in Figure 2, the **GPU is by far the primary consumer of energy** throughout all of the stages of the inference process. In contrast, CPU and RAM usage contribute only marginally to the total energy budget.

When analyzing the inference process, the prefill phase incurs the highest computational cost relative to the number of output tokens generated. This is because the model processes the entire input sequence and computes key/value caches for all input tokens before generating the first token. While computationally expensive, this step remains highly parallelizable on the GPU, which helps mitigate its efficiency.

The *decode* phase is inherently **autoregressive** and therefore less parallelizable: each new token depends on the previous one and reuses the cached context without recomputing the entire key/value cache. As a result, each decode step is lighter in isolation but must be repeated sequentially for each token generated. This sequential nature keeps the GPU occupied for a longer time, leading to higher total energy for the decode step, especially for longer output lengths. Figure 3 highlights this: while prefill consumes a significant chunk of energy up front, the long tail of high energy usage comes mainly from the repeated decode operations that accumulate over long outputs.

---

[2]The "±" symbol denotes the variance in energy usage across generations in the dataset

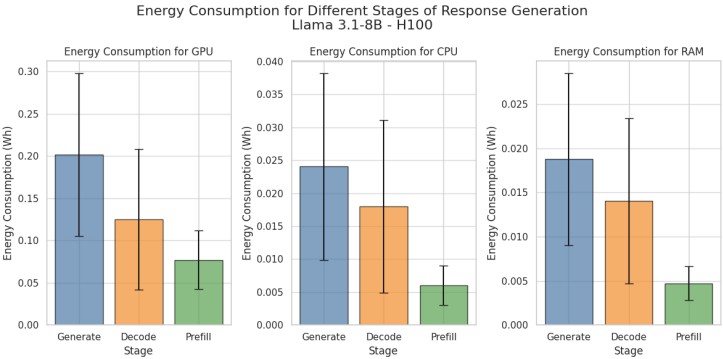

Figure 2: Energy consumption by hardware type (GPU, CPU, RAM) and phase (prefill, decode, generate). The GPU consistently dominates across all phases, while CPU and RAM consumption remains minor.

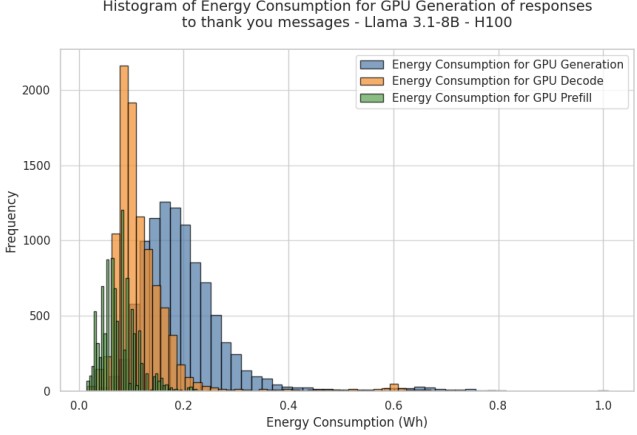

Figure 3: Histogram of GPU energy consumption for prefill and decode phases. Decode contributes most to the long tail due to repeated sequential steps, whereas prefill energy is substantial but occurs only once per request.

## 5 ENERGY AND LATENCY DEPENDENCE ON INPUT AND OUTPUT LENGTH

### 5.1 THEORETICAL LATENCY MODEL

To better understand how input and output lengths influence energy consumption, we adopt a closed-form latency model based on the dominant operations during LLM inference. Each GPU kernel is modeled as either compute-bound or memory-bound, depending on its floating-point operation count $F_o$ and data transfer volume $D_o$, relative to hardware ceilings: floating-point throughput $F_{\max}$ and memory bandwidth $B_{\max}$. The theorical effective latency of an operation is:

$$t_o = \max\left(\frac{F_o}{F_{\max}}, \frac{D_o}{B_{\max}}\right),$$

The total latency is the sum over all operations:

$$T = \sum_o t_o.$$

An operation is *compute-bound* when its execution time is limited by arithmetic throughput, and *memory-bound* when data movement dominates. On modern GPUs, compute and memory operations can often proceed asynchronously, competing for shared resources like streaming multiprocessors

(SMs) or memory buses. Our model conservatively assumes no overlap, thus upper-bounding the true latency.

We neglect kernel launch overhead, although it can become significant when GPU kernels execute faster than the CPU can dispatch them - especially in regimes with short or tightly chained kernels (e.g., decode or layernorm blocks), where inter-kernel gaps due to CPU scheduling latency inflate total latency. Additionally, actual data transfer speeds can vary depending on caching effects - for instance, if intermediate activations are reused and remain in L2 or shared memory, memory-bound operations may be accelerated.

To account for these approximations, we introduce empirical efficiency factors for compute and memory:

$$F_{\text{eff}} = \mu_{\text{comp}} \cdot F_{\text{max}}, \quad B_{\text{eff}} = \mu_{\text{mem}} \cdot B_{\text{max}},$$

where $\mu_{\text{comp}}$ and $\mu_{\text{mem}}$ absorb multiple sources of inefficiency. These include suboptimal GPU occupancy, memory misalignment penalties, limited overlap between data fetch and compute, and variable cache residency for activations.

The values of $\mu_{\text{comp}} = 0.675$ and $\mu_{\text{mem}} = 0.443$ used throughout this section were calibrated empirically using profiling data from LLaMA 3.1–8B (in FP32 precision) inference on an NVIDIA H100 SXM GPU.

**Prefill phase.** This phase processes the prompt of length $s$ through $N = 32$ transformer blocks. The dominant operations are compute bound for input sequence greater than 100 and include QKV projections, feed-forward layers and FlashAttention. Using the theoretical model described above, we approximate the latency of the prefill phase with the following fitted expression:

$$t_{\text{prefill}}(s) \approx \alpha s + \beta s^2 + \gamma,$$

$$\alpha \approx 3.18 \times 10^{-4} \text{ s/token}, \quad \beta \approx 1.17 \times 10^{-8} \text{ s/token}^2.$$

$$\gamma \approx 1.68 \times 10^{-2} \text{ s}$$

In practice, latency is constant for very short prompts ($s \lesssim 100$), linear across most real-world prompts ($s < 4{,}000$), and quadratic only for extreme lengths ($s > 30{,}000$).

**Decode phase.** This phase generates $g$ tokens autoregressively, attending to a growing context $\ell = s + t - 1$. All operations remain memory-bound. Total latency scales as:

$$t_{\text{decode}}(s, g) \approx \eta g + \theta s g + \phi g^2 + \rho,$$

$$\eta \approx 2.61 \times 10^{-2} \text{ s/token}, \quad \theta \approx 3.31 \times 10^{-7} \text{ s/token}^2.$$

$$\phi \approx 5.86 \times 10^{-8} \text{ s/token}^2, \quad \rho \approx -5.32 \times 10^{-2} \text{ s}.$$

Quadratic effects in $g$ exist theoretically but appear only for $g \gtrsim 10^5$, beyond practical usage.

## 5.2 Link to Empirical Energy Trends

The latency trends outlined above align closely with our energy measurements. On the H100 GPU, we can record the average power during each phase - 684 W during prefill (for average $s$) and 293 W during decode (for average $s$ and batch size of 1). As a result, energy consumption becomes approximately proportional to runtime, with an effective power $\bar{P}_{\text{eff}}$ that remains nearly constant within each phase. This proportionality justifies the direct mapping between theoretical latency and empirical energy trends.

Figure 4 illustrates the observed relationship: energy in the prefill phase scales linearly with input length, while decode energy grows primarily with output length.

**Prefill energy.** Energy in the prefill phase increases linearly with the number of input tokens:

$$E_{\text{prefill}}(s) \approx A \cdot s + B,$$

with fitted values:

$$A \approx 6.05 \times 10^{-5} \text{ Wh/token}, \quad B \approx 5.00 \times 10^{-3} \text{ Wh}.$$

This is consistent with the compute-bound regime described in the latency model.

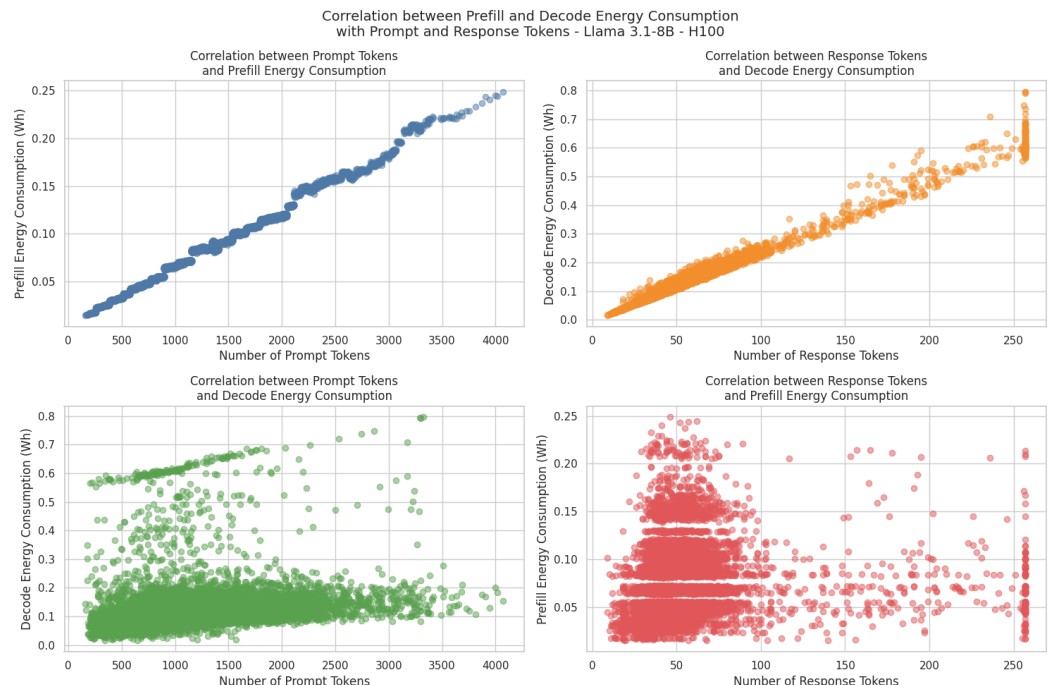

Figure 4: Correlation between token lengths and GPU energy consumption in prefill and decode phases.

**Decode energy.** For generation lengths $g << 10^5$, decode energy follows the linear regime predicted by:

$$E_{\text{decode}}(s, g) \approx Cg + Dsg + G$$

with fitted values:

$$C \approx 2.13 \times 10^{-3} \text{ Wh/token} \quad D \approx 2.87 \times 10^{-7} \text{ Wh/token}^2.$$

$$G \approx -4.71 \times 10^{-3} \text{ Wh}$$

This reflects the cumulative cost of attending to a growing context at each step. The weak dependence on $s$ corresponds to the repeated attention over the prompt tokens during generation.

**No quadratic growth.** While the theoretical model includes a quadratic term in $g$, this component remains negligible in practice given that our dataset does not include sequences long enough ($g \gtrsim 10^5$) to observe this behavior.

**Kernel effects.** Discrete jumps and non-linearities in the measured energy arise from low-level kernel effects such as block alignment and tiling. These artifacts do not contradict the overall linear trends and are typical of GPU workloads (see bottom right subplot on Figure 4).

**Summary.** Energy during inference scales linearly with prompt and generation lengths, matching the theoretical latency model under stable power conditions. The decode phase shows a bilinear dependency, with output length as the dominant factor and a minor prompt-length contribution.

## 6 IMPACT OF MODEL SIZE ON ENERGY CONSUMPTION

To assess how model scale influences energy usage, we extended our analysis beyond LLaMA 3–8B to include models from the **Qwen 2.5 family** (ranging from 0.5B to 14B parameters) and **Mistral–7B**. Since Qwen models share the same architecture and tokenizer, this allowed for a controlled comparison focused solely on model size.

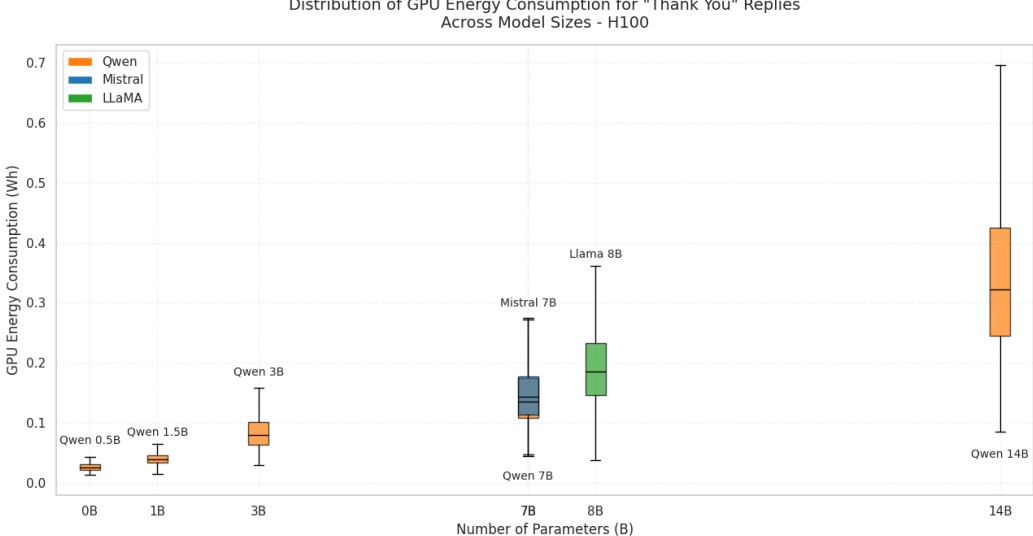

Figure 5: GPU energy consumption during generation as a function of model size. Boxes represent the distribution across 10,000 replies including polite "thank you" phrases.

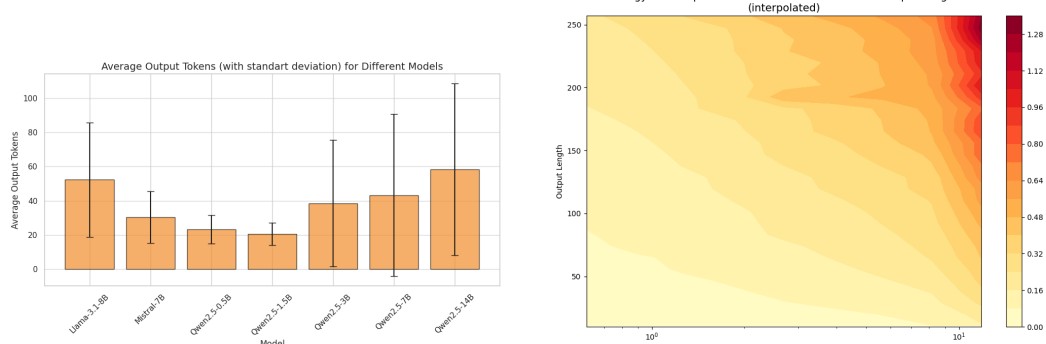

(a) Mean and standard deviation of output token lengths across models. Larger models tend to generate longer responses on average.

(b) Interpolated contour plot of GPU energy consumption (Wh) during the decode phase, as a function of model size (log scale) and output length.

Figure 6: (Left) Output length increases with model size; (Right) decode energy depends on both output length and model size.

We observed that **larger models have the tendency to produce longer outputs** on average (Figure 6a). This reflects their greater capacity to elaborate responses, but also contributes to increased compute and energy during inference.

As shown in Figure 5, energy usage scales with model size – while smaller models are more energy-efficient, larger models incur significantly higher costs to generate responses of similar or slightly longer lengths.

To isolate the effects of model size and verbosity, we analyzed decode-phase energy as a function of both parameters. Figure 6b shows that **model size is the dominant factor**, with energy increasing steeply along the model axis. Output length has a secondary influence but remains within a narrower energy band.

**Link to theoretical latency analysis.** The observed increase in energy with model size can be directly explained by the architectural scaling effects. Specifically:

- The number of transformer blocks $N$ grows with model size and contributes **linearly to the total latency** in both prefill and decode phases.

- In the prefill phase, dominant operations such as QKV projection and feedforward layers involve matrix multiplications with complexity $\mathcal{O}(sNh^2)$, leading to a **quadratic dependence on the hidden dimension** $h$.

- In the decode phase, energy scales as $\mathcal{O}(gNh^2)$ in the memory-bound regime

Together, these explain why energy increases steeply with model size: larger models have deeper networks ($N$), wider layers ($h$), and tend to generate longer outputs ($g$). This is reflected in the contour plot of Figure 6b, where energy increases most rapidly along the model-size axis.

**In summary**, scaling up model size increases energy use significantly - not only due to longer replies, but also due to larger hidden dimensions and more layers. These results emphasize the importance of considering model size–efficiency trade-offs, even in seemingly trivial interactions such as generating an output in response to a "thank you". Although larger models consume more energy, their outputs may exhibit higher helpfulness, informativeness, or linguistic fluency - aspects which we do not evaluate in this study.

## 7  RELATED WORK

**Energy Cost of LLM Inference.**   As LLMs grow in usage and scale, inference has become a dominant source of energy consumption in AI deployments (Luccioni et al., 2024; 2022). Early studies emphasized training costs (Strubell et al., 2019; Patterson et al., 2021; Henderson et al., 2020), but recent work highlights the environmental impact of inference across real-world workloads (Fernandez et al., 2025). Many strategies have been proposed to improve inference efficiency, including serving configurations (Face, 2022; NVIDIA, 2023; Dao et al., 2022), batching techniques (Kwon et al., 2023; Yu et al., 2022), and decoding strategies (Ding et al., 2024; Leviathan et al., 2023). While prior efforts often rely on theoretical FLOPs or GPU utilization, empirical results show large gaps between these and actual energy use. Broader perspectives advocate for efficiency as a core research metric (Schwartz et al., 2019), standardized reporting (Luccioni et al., 2024; Tschand et al., 2025), improved cost indicators (Dehghani et al., 2021) and for accounting emissions at both system and hardware lifecycle levels (Wu et al., 2021). Despite this, few studies focus on the energy cost of micro-interactions, such as polite phrases like "thank you," which may appear negligible but compound significantly at scale.

**Prompt and Interaction Length.**   Prompt verbosity significantly influences LLM inference efficiency, both in latency and energy. Adamska et al. (2025) introduced "Green Prompting," showing that semantic density and prompt length impact energy use. Poddar et al. (2025) demonstrated that minimizing output verbosity can yield up to 60% energy savings. Gao et al. (2024) extended this idea to vision-language models, where longer generations induced by "verbose images" substantially increased latency and energy. Other studies (Fernandez et al., 2025; Wilkins et al., 2024; Patel et al., 2024; Jin et al., 2025) highlight how prompt/output length shapes real-world workload geometry, affecting batching, memory use, and phase-level energy patterns. Building on these, we show that even short polite phrases like "thank you" can increase both prompt and response lengths, subtly yet consistently raising energy costs in conversational LLM use.

**Politeness and User Behavior.**   Although politeness in human-computer interaction is a well-studied phenomenon (Chow, 2023), there is little work on its specific energy implications in the context of LLMs. The "ELIZA effect," wherein users anthropomorphize early AI systems, continues to persist with modern LLMs (Hofstadter, 1995). Recent studies have begun to investigate the influence of polite prompts on model behavior (Yin et al., 2024), showing that such linguistic choices can affect LLM outputs across languages. While these behaviors play important social roles, they also tend to increase prompt length and output verbosity, leading to higher energy costs, an aspect that has not been systematically quantified in prior work.

**Energy Measurement Tools.**   Reliable energy measurement is essential for evaluating the efficiency of LLMs. Tools like CodeCarbon (Courty et al., 2024), pyRAPL (pyRAPL contributors, 2020), and NVIDIA's NVML enable detailed tracking of GPU and CPU energy usage during inference. In

this work, we adopt these frameworks to measure the energy consumption of LLM inference in fine-grained detail, focusing specifically on polite prompts such as "thank you."

**Summary.** Past work has improved LLM inference efficiency via prompt tuning, batching, and decoding strategies, but rarely examines ubiquitous micro-interactions like "thank you." We fill this gap by quantifying their energy cost across models and phases, and linking it to a closed-form latency model capturing compute- vs memory-bound regimes. This bridges behavioral and system-level perspectives, introducing polite prompts as a reproducible unit for fine-grained energy analysis.

## 8 LIMITATIONS AND FUTURE WORK

This study focuses on politeness - and in particular the ubiquitous "thank you" message - as a controlled unit of interaction to explore the energy dynamics of LLM inference. While this setup allows reproducibility and stable comparisons, several limitations remain:

- **Prompt and output diversity.** Our dataset contains short, polite prompts and uses batch size of 1. Future work should explore longer dialogues, larger batches, and more complex tasks to capture non-linear behaviors and scaling effects.
- **Model and backend scope.** We focused on a limited set of open-source LLMs and used PyTorch inference on a single H100 GPU. Expanding to other hardware (e.g., A100, L4, AMD GPUs, TPUs) or optimized runtimes (e.g., vLLM, TGI) would allow broader generalization.
- **Latency model simplifications.** Our closed-form model abstracts away kernel launch overhead, cache-level reuse, and dynamic power variation. A more fine-grained analysis could incorporate kernel profiling, hardware-level instrumentation, and study how GPU power evolves with input size, batch size, and workload intensity.
- **From energy to impact.** We measured raw energy in Wh, but did not convert to carbon emissions or dollar cost. Future work could map these energy values to environmental and economic externalities, and incorporate accuracy trade-offs in real-world applications.

## 9 CONCLUSION AND TAKEAWAYS

We used the "thank you" prompt as a reproducible micro-interaction to study the energy profile of LLM inference. This seemingly trivial interaction exposes clear structural trends in energy usage - and reflects broader trade-offs in sustainable AI deployment.

Our main takeaways are the following:

- **Input/output length matters.** Energy grows linearly with prompt and generation length. Decode dominates for long outputs due to its sequential nature.
- **Model size matters.** Larger models produce longer replies and incur steeper energy costs - both due to deeper architectures and longer inference runtimes.
- **Phase distinction is key.** Prefill is compute-bound and benefits from hardware parallelism; decode is memory-bound and more sensitive to sequential overhead.
- **Latency models are useful.** Our closed-form model aligns with empirical trends, and offers a foundation for predicting inference energy at scale.

In conclusion, politeness isn't free - but understanding its cost helps build more efficient and sustainable LLM deployments. By quantifying where energy is spent, we can begin to optimize not just model performance, but also its environmental footprint.

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
