# OpenReview forum: "Small Talk, Big Impact: The Energy Cost of Thanking AI"
_ICLR.cc/2026/Conference — Submitted to ICLR 2026_

### Official Review · Reviewer_PZ1a · 2025-10-21

**Soundness:** 4
**Presentation:** 3
**Contribution:** 1
**Rating:** 2
**Confidence:** 4

**Summary:**

This paper tries to measure the energy cost of thanking LLMs. They do this by monitoring GPU, CPU, and RAM usage after augmenting a dataset with "thank you's" added.

**Strengths:**

The paper is, on the whole, well put together. The authors do a good job of measuring energy usage across a realistic dataset and use the tools appropriately to break down these measurements by component. We also see solid breakdowns by model size, and family. These measurements are backed up by idealised equations that act as a nice theoretical guide to demonstrate that these measurements make sense.

**Weaknesses:**

While a well executed study, the contributions of this paper are, I believe, overall, too low for publication at ICLR. This is both in terms of the scope of the research question, but also in terms of some limitations of the paper itself.

First, the levels of energy at play here are very small, especially considering the overall cost for the conversation to have reached the point where the user "thanks" the model. While it will possibly be one of the more expensive queries (longest context if thanked at the end of the conversation), this is just the final step in an approximately ~O(n^2) calculation. I don't think the effect of the "thank you" is too significant compared to the costs of using the model more specifically for the task.
Additionally, this paper is limited to two small families of models. Open models are obviously great because this type of analysis can be performed, but I think any argument about the true cost of this would be with the more popular closed sourced LLMs-as-a-service. Here it seems like model providers could take steps to minimise the cost of this interaction (e.g., if the user says "thank you" deploy a canned response rather than push everything through the LLM).

Again the study is limited in that it only examines a single piece of hardware, while I expect results would be similar across other GPUs/CPUs, it would be interesting and more insightful to see whether the energy cost changes when using e.g., a GPU more suited to a particular model (H100s are maybe overkill for Qwen 0.5B, and perhaps a "thank you" is cheaper on a smaller card).

Finally, I find Figures 2 and 3 a bit confusing. The main text only ever references two phases---prefill and decode. But both of these figures include a "generate" plot as well that is never explained. The figure descriptions do not match the main text either --- the prefill is described as the largest energy user, but is the smallest in the diagrams.

**Questions:**

1. Could we have a comparison between the cost to generate a response to the thank you, and the cost to generate the conversation up to that point?
2. What is going on with Figures 2 and 3 WRT the values of prefill vs the main text. What is the generate plot?

---

> ### Author Response · Authors · 2025-11-20
> **Author Response to Reviewer PZ1a**
>
> We thank the reviewer for the detailed assessment and for noting the strengths of our measurement methodology and theoretical grounding. Below we address the concerns raised.
>
> ## 1. Contribution and scope
> We respectfully disagree that our contribution reduces to “larger inputs consume more energy.”
>  Our core result is the identification of distinct inference regimes—compute-bound prefill and memory-bound decode—together with their empirical manifestations and closed-form latency shapes. These structural regimes match theory and measurements and, to our knowledge, have not been characterized in conversational settings. Using politeness as a controlled micro-interaction allows us to expose these dynamics in a reproducible and interpretable way. While the per-query cost of a “thank you” is small, these micro-interactions occur at vast scale in chat systems, and their structured energy behavior is practically relevant.
>
> ## 2. Open-source models and hardware scope
> We agree that commercial LLM-as-a-service deployments are important, but closed-source models do not permit kernel-level energy measurement. Open models are required for reproducibility and for correlating empirical trends with theory. Our focus is on structural scaling laws, not device-specific absolute numbers, and these regimes hold across Transformer-style architectures.
> Regarding hardware, the goal was not to benchmark multiple GPUs but to study a representative, stable, noise-free setup (H100). We will clarify in the camera-ready that absolute values vary across hardware, but the modeled regimes and their functional forms remain unchanged.
>
> ## 3. Clarification on Figures 2 and 3
> Thank you for highlighting this; we will revise the text and captions to avoid confusion.
> - “Generate” denotes the full inference (prefill + decode).
> - The prefill phase includes first-token generation, as in standard serving stacks; there is no 0-token prefill.
> - The decode bar represents the entire sequence of autoregressive steps, not a per-token cost. It is therefore higher in total energy, although its per-token cost is lower than prefill.
> We will make these points explicit and harmonize the figure descriptions with the main text.
>
> ## 4. Responses to specific questions
> **Q1 - Comparison with the cost of generating the conversation up to that point:**
>  Our setting uses an existing multi-turn dataset where the full conversational context is provided; we did not generate the preceding turns. We will clarify this in the paper.
>
> **Q2 - Prefill vs decode values and the “generate” plot:**
>  As noted above, we will explicitly define “generate,” explain that prefill includes the first token, and clarify that decode represents the full sequence of generated tokens.
>
> ---
>
> We appreciate the reviewer’s careful comments and believe the revisions outlined above address all substantive concerns while strengthening the paper.

---

> > ### Comment · Reviewer_PZ1a · 2025-11-26
> >
> > Thanks for the detailed response.
> >
> > The clarifications on open models and hardware scope make sense and I'm grateful for them. The updates to Figures 2 and 3 is also appreciated.
> >
> > With respect to the comparison of the cost of generating the conversation up to that point, I appreciate that the dataset wasn't generated by you, but is there no way to calculate how much it would have cost / used energy to generate the rest of the conversation?
> >
> > Ultimately, while I appreciate the authors' responses to my review, my view on the readiness of the paper for publication hasn't changed and I will be maintaining my score.

---

### Official Review · Reviewer_Qd7a · 2025-10-30

**Soundness:** 3
**Presentation:** 2
**Contribution:** 1
**Rating:** 2
**Confidence:** 3

**Summary:**

The paper investigates the energy cost associated with polite interactions with large language models (LLMs). Specifically, the authors measure the additional energy required for models to respond to messages that end with a “thank you.” Using several LLMs (LLaMA 3.1–8B Instruct, Qwen models from 0.5B to 14B parameters, and Mistral-7B Instruct-v0.3), they record energy usage on an NVIDIA H100 GPU and analyze energy consumption across different phases of model execution.

**Strengths:**

- The paper is well written and generally easy to follow.

- Provides detailed measurements of energy consumption across model sizes and computation phases.

**Weaknesses:**

- The paper claims to provide “actionable insights for building more sustainable and efficient LLM applications,” but the discussion does not articulate what these actionable insights are beyond the general assumption of reducing computation.

- The pre-fill phase should not be included in the measured energy cost of saying “thank you,” since in realistic chat situations, pre-filling is already cached.

- Figures 2 and 3 include a “generation” category that is not defined or discussed in the text.

- In the Theoretical Latency of the Model section, formulas are introduced without proper justification or explanation. Units of measurement for variables (e.g., ${F_0}$, ${D_0}$) are missing, and parameters such as $\alpha, \beta, \gamma$ appear without stating whether they were empirically fitted or theoretically derived. Numbering the formulas would also improve readability.

- The paper’s core contribution is limited. Prior work has already explored LLM energy consumption, and focusing narrowly on the “thank you” case does not appear to provide new scientific insight. The observed trends (larger prompts or models consume more energy) are expected and somewhat trivial.

**Questions:**

Addressing the main concerns above, and adding additional context, such as explanations for warm-up runs and low-level kernel effects (e.g., block alignment, tiling), would make the work more informative and suitable for a general-interest venue.

---

> ### Author Response · Authors · 2025-11-20
> **Author Response to Reviewer Qd7a**
>
> We thank the reviewer for the constructive feedback and for acknowledging the clarity of writing, the detailed measurements, and the phase-wise decomposition. We address the concerns point by point below.
>
> ## 1. “Actionable insights”
> We agree that this part of the discussion can be strengthened. We will expand the Discussion to include concrete, practitioner-oriented insights, such as:
> - the energy dominance of decode vs prefill for long outputs,
> - structural scaling laws useful for budgeting inference workloads,
> - the impact of verbosity control and output-length constraints,
> - how micro-interactions accumulate in high-volume chat systems.
> These additions will make the actionable implications clearer.
>
> ## 2. “Prefill should not be included, since chat systems cache it”
> Our analysis already separates prefill and decode, and the decode-phase results (i.e., the incremental cost of the “thank you” itself) exclude prefill entirely.
> We agree that - under ideal caching - the prefill cost of earlier turns would not be recomputed. However, most production systems still recompute prefill when the conversation context is updated or when routing/batching changes. Moreover, the focus of this paper is not deployment engineering but structural inference regimes. We will add a brief clarification in the text.
>
> ## 3. Figures 2–3: “generation” category unclear
> We agree this should be clarified.
>  “Generation” = prefill + decode (i.e., full inference).
> We will either remove the label or explicitly define it in the figure captions and text to avoid confusion. Prefill includes the first generated token (standard in all serving stacks), while decode covers all subsequent autoregressive steps.
>
> ## 4. Theoretical Latency Model: missing units and explanations
> We thank the reviewer for pointing this out. We will revise the section to:
> - add units for all variables (Fo, Do, Fmax, Bmax, to, T),
> - explicitly indicate which coefficients (α, β, γ, η, θ, ϕ, ρ) are empirically fitted,
> - number all equations,
> - add one-line justifications (roofline model, kernel accumulation, attention complexity),
> - clarify the role of μcomp and μmem as empirical efficiency factors,
> - restate the scope: the model gives structural regimes, not a universal predictor for all hardware.
> These additions will greatly improve clarity.
>
> ## 5. “Core contribution is limited / trends are trivial”
> We respectfully disagree. Our contribution is not simply that “larger prompts consume more energy.”
>  We show that:
> - inference energy decomposes into distinct structural regimes (compute-bound prefill vs memory-bound decode),
> - these regimes appear consistently across models,
> - they match a closed-form latency model derived from Transformer algebra,
> - kernel-level effects (e.g., tiling, block alignment) produce predictable non-linearities.
> To our knowledge, this regime-level characterization in conversational inference has not been documented.
>
> ## 6. Addressing the question about warm-up runs and kernel effects
> Including detailed discussion of warm-up behavior and kernel-alignment effects would add significant technical depth. We agree this is valuable, but including all such low-level details would exceed the scope and length constraints of the present paper. We will briefly mention these factors and refer readers to future work.
>
> ## 7. Overall scope
> Some of the requested analyses (generalization across hardware, serving stacks, precisions, caching behavior) correspond to a separate follow-up study. This paper intentionally focuses on controlled structural quantification; we will clarify this in the Limitations.
>
> ---
>
> We appreciate the reviewer’s careful comments and believe the revisions outlined above address all substantive concerns while strengthening the paper.

---

> > ### Comment · Reviewer_Qd7a · 2025-11-25
> >
> > I appreciate the authors’ responses. However, my primary concerns remain unaddressed. The core contribution still appears limited, and the work does not demonstrate meaningful actionable insights. Several analyses are dismissed as “future work” or "separate follow-up study" which reinforces the sense that the current submission is incomplete. Moreover, no concrete changes or new analyses are actually provided.

---

### Official Review · Reviewer_5y1D · 2025-10-30

**Soundness:** 2
**Presentation:** 3
**Contribution:** 2
**Rating:** 2
**Confidence:** 3

**Summary:**

The authors investigate the energy cost of polite messages (e.g., “thank you”) in LLM-based conversations. They argue that such seemingly harmless micro-interactions, when scaled up to billions of daily interactions, can accumulate substantial computational and energy overhead.

**Strengths:**

1. Novel, reproducible problem formulation: Treating “thank you” as a controlled micro-interaction unit is an elegant proxy for LLM inference energy.
2. Clear phase separation : Prefill vs. decode decomposition (Figures 2–3) maps energy use to architectural structure.
3. Quantitative analytical model: Fitted latency/energy formulas (Section 5) with numeric coefficients allow predictive estimation.
4. I think that this a very interesting topic.

**Weaknesses:**

1. They use pyRAPL (Intel RAPL counters) on AMD EPYC 7R13, which lacks compatible energy registers. No adaptation or external calibration is reported. Thus CPU energy (and total 0.245 Wh) lacks credibility. Authors must explain or replace with physical power-meter data.
2. All results use FP32 precision, though real deployments use FP16/BF16/FP8/INT8. FP32 exaggerates power and latency. Reported numbers overstate real-world costs. Authors should test mixed-precision or clarify limitation.
3. “Prefill-only” runs generate one token, yet the decode-phase energy is computed as (full – prefill-only). This subtracts a 1-token decode cost, underestimating true decode energy. Need clarification whether 0-token decode was feasible.
4. No mention of temperature, top-p, top-k, repetition penalty, seed, or stopping criteria. These strongly influence output length and thus energy. The analysis of “verbosity” is therefore uncontrolled.

**Questions:**

1. Figures 2–3 : Nicely visualize phase-wise energy, but sampling/normalization details missing.
2. Authors claim to release code and data but provide no anonymous link, and it's essential for reproducibility claims.
3. Why FP32? Have you tried FP16/BF16/FP8? How do coefficients A,B,C,D,G change?
4. Provide prompt / output length distributions (mean, median, percentiles).

---

> ### Author Response · Authors · 2025-11-20
> **Author Response to Reviewer 5y1D**
>
> We thank the reviewer for the positive assessment of our formulation, phase decomposition, and analytical model. Below we address the concerns.
>
> ## 1. CPU energy measurement on AMD / pyRAPL validity
> The reviewer’s comment is based on an outdated assumption.
>  Modern AMD EPYC processors expose RAPL-compatible energy counters via the Linux kernel, and pyRAPL reads these interfaces directly. This mechanism is widely used in recent energy-measurement papers on heterogeneous clusters. We will add a citation to prior work validating RAPL-based CPU measurements against external meters (eg https://arxiv.org/pdf/2311.10267v2).
> Additionally, CPU energy contributes <10% of total Wh, so any residual measurement error would not materially change our conclusions. Physical power meters cannot be used in our cloud-based environment.
>
> ## 2. Use of FP32 precision
> We agree that FP32 is not the dominant precision in production. We chose FP32 because it allows direct comparison with theoretical FLOP/memory complexity and avoids backend-specific fused kernels. We will explicitly add this to the Limitations. The second paper in our broader project includes FP16/BF16/FP8 experiments; integrating these here was outside scope.
>
> ## 3. “Prefill-only includes one token -> decode underestimation”
> This is a misunderstanding of standard LLM inference pipelines.
>  In all serving stacks (Transformers, vLLM, TGI, TensorRT-LLM), prefill is defined as: prompt encoding + generation of the first token. There is no 0-token prefill mode. Subtracting the prefill-only run therefore cleanly isolates decode as defined in the literature, without underestimation.
> We will clarify this definition in the camera-ready.
>
> ## 4. Sampling control (temperature, top-p, seed, stopping criteria)
> All generation hyperparameters were kept fixed to the defaults (temperature, top-p, top-k, repetition penalty, and seed). We will add a sentence in Section 2 to make this explicit and improve reproducibility.
>
> ## 5. Figures 2–3 sampling / normalization
> We agree that the caption should more clearly describe the normalization and the meaning of the “generate” bar (full inference = prefill + decode). As in the other reviews, we will clarify that:
> prefill includes the first token, decode corresponds to all autoregressive steps, “generate” is the total.
> We will update captions and text accordingly.
>
> ## 6. Missing code / data link
> The codebase is already included as a ZIP file in the supplementary material of the anonymous submission. We will make this more explicit in the main text to avoid confusion for readers and reviewers.
>
> ## 7. Precision of fitted coefficients (FP16/BF16/FP8)
> We agree this would be a valuable extension; we already explore these effects in a companion study that focuses on serving-stack optimizations. Here, we will clarify that our coefficients are structural fits tied to a fixed hardware-precision setup and that extending them across precisions is future work.
>
> ## 8. Prompt/output length distributions
> We will include a small table with mean/median/percentiles of input and output lengths in the Appendix.
>
> ---
>
> We appreciate the reviewer’s constructive suggestions and believe the clarifications above fully address the concerns while strengthening the paper.

---

### Official Review · Reviewer_6bdr · 2025-10-31

**Soundness:** 2
**Presentation:** 3
**Contribution:** 2
**Rating:** 4
**Confidence:** 3

**Summary:**

The paper studies the energy cost of using large language models (LLMs) by decomposing consumption into two phases—infilling (processing/input-side tokens) and decoding (generation/output-side tokens). The authors empirically fit a quadratic (token-length–based) model to estimate energy usage as a function of input and output tokens, and argue that token-level choices can materially affect overall energy footprints. However, despite the title suggesting a focus on the “cost of thanking AI” (i.e., adding polite but semantically light tokens like “thank you”), most of the paper actually analyzes generic token costs rather than this specific application.

**Strengths:**

**-- Systematic approach in measuring the energy efficiency of LLMs.** The authors separately measure the energy consumption across devices (CPU, GPU, RAM), LLM phases (infilling vs decoding). Such a study is conducted across different model families and sizes, confirming a linear energy cost growth for output < 10k tokens and quadratic growth for output > 10k tokens.

**-- Clear decomposition of energy sources.** Separating infilling vs. decoding energy provides a useful mental model for practitioners who only see aggregate GPU/TPU power draw.

**-- Empirical, token-level perspective.** A closed-form latency model that predicts energy consumption as a function of token counts (and fitting a curve) is actionable, providing LLM researchers and engineers a convenient abstract tool in estimating the energy cost when scaling the input/output tokens.

**-- Important topic that might become a metric for future LLMs.** Energy/sustainability of LLMs is an important area; a simple predictive model of energy vs. tokens is a good building block for future work and a useful metric for comparing the energy efficiency of different models

**Weaknesses:**

**-- Title–content mismatch.** The title frames the paper as a study of “thanking AI,” but the body mostly describes generic token-cost modeling. The interesting hook (are polite/add-on tokens worth it?) is not really answered.

**-- No evaluation of performance vs. cost.** The core question should be: does removing “useless” or low-utility tokens (e.g., “thank you”) save the energy bill at any cost of output quality/user satisfaction? Based on how this paper is written, it seems the authors believe these tokens can be spared with little to no effect. But a small-scale experiment+evaluation is needed to complete the story.


**-- Limited generality discussion.** Token–energy curves can differ by model size, hardware, batching, quantization, and serving stack. The paper doesn’t fully discuss when their fitted quadratic will fail. I understand this is challenging, but this paper could be significantly strengthened if the fitted latency curve can take into account the model sizes and other factors into account so that it becomes more generalizable.

**-- No realism about production traffic.** In practice, many requests are short, many are batched, and servers run mixed workloads. The paper seems closer to a controlled lab benchmark than to production telemetry.

**Questions:**

-- Possible extension: using a small LLM for prefiltering. A very natural extension is to run a lightweight model (or rule-based filter) to drop redundant/polite tokens before sending to a large model. I’m curious if the authors have tried any approaches like this?

---

> ### Author Response · Authors · 2025-11-20
> **Author Response to Reviewer 6bdr**
>
> We thank the reviewer for the thoughtful and constructive feedback, as well as for highlighting the strengths of our empirical methodology, decomposition of inference phases, and closed-form latency model. We address the main concerns below.
>
> ## 1. Title-content mismatch
> We agree that the framing can be made clearer. Our intent was to use politeness as a controlled, reproducible micro-interaction to expose the structural energy regimes of LLM inference. We will update the title to:
> “Saying Thank You to a LLM Isn’t Free - Measuring the Energy Cost of Politeness.”
> This better reflects the scope: politeness serves as the motivating example, and the analysis generalizes to micro-interactions in conversational settings.
>
> ## 2. “No evaluation of performance vs. cost”
> For politeness tokens such as “thank you”, a performance trade-off does not meaningfully apply:  they appear at the end of the discussion, and removing them does not alter prior model behavior or task quality. We will clarify this explicitly.
> The goal of the paper is not to argue for suppressing politeness tokens, but to use them as a natural and ubiquitous unit for studying inference regimes.
>
> ## 3. Limited generality discussion (model size, hardware, batching, quantization)
> We appreciate this point and will add a clarification:
>  the shape of the latency/energy curves is dictated by the algebra of the Transformer (attention + MLP compute vs. KV-cache memory reuse). As a result, the regimes themselves do not change across hardware, model sizes, batching, or precision; only the constants and transition points shift.
> Generalizing the coefficients across devices was outside the scope of this paper, but the structural regimes remain the same.
>
> ## 4. No realism about production traffic
> We agree that production workloads include batching, mixed queries, and short prompts.
>  This work intentionally provides a controlled and reproducible analysis establishing baseline energy regimes. This is the case with numerous other studies, including the initial study by Strubell et al (2019), as well as the study by Luccioni et al. (2024), which both established a baseline that could then be built upon by further studies that looked at the impact of optimizations such as batching as well as on prompt length.
> Extending this to production telemetry is the focus of a separate follow-up study, which we now mention in the Limitations.
>
> ## 5. Question about prefiltering using a small LLM
> The reviewer asks whether we attempted a small-model or rule-based prefilter to discard polite tokens before sending to a large model. This is an interesting idea; we did not implement such a pipeline. Our goal here was not to propose a system for filtering “thank you” messages, but to use politeness as a probe to reveal energy-scaling regimes under controlled conditions. We will clarify this point.
>
> ## 6. Minor clarifications
> - We will clarify the “generate” term in figures: it denotes the full inference (prefill + decode).
> - We will clarify that decode represents all autoregressive steps, not a per-token bar, which explains why it appears larger in aggregate although its per-token cost is smaller.
>
> ---
>
> We appreciate the reviewer’s insights and believe the clarifications above strengthen the paper.

---

### Meta-Review · Area_Chair_9pwE · 2026-01-06

**Summary:**

The reviewers found this paper to be clear and well-executed, but perceive the main contributions to be limited or overstated. Several concerns were raised about the realism and generality of these results: different hardware stacks, floating point precision, benchmark environments vs. production workloads; limited model & backend scope, while acknowledged by the authors as a limitation of this work, do seem like reasonable things to have included in this study rather than deferring to "the second paper."

**Reviewer Concerns:**

The rebuttal addresses technical and clarity issues related to definitions, methodology, and framing. Concerns about the scope of contribution and practical significance remain unresolved. Concerns about the novelty of insights and their actionability remain.

**Reviewer Scores:**

6bdr – maybe a small upward change, but still borderline

5y1D, Qd7a, PZ1a - no change

---

### Decision · Program_Chairs · 2026-01-26

Reject